# Trajectory Planning for Spray Painting Robot Based on Point Cloud Slicing Technique

**Wei Chen [1,2,*], Xu Li [1], Huilin Ge [1] , Lei Wang [1] and Yuhang Zhang [1]**

[1] School of Electronics and Information, Jiangsu University of Science and Technology, Zhenjiang 212003, China; 189030040@stu.just.edu.cn (X.L.); ghl1989@just.edu.cn (H.G.); 189030031@stu.just.edu.cn (L.W.); 189030046@stu.just.edu.cn (Y.Z.)

[2] School of Automation, Southeast University, Nanjing 210096, China

* Correspondence: cw1@just.edu.cn

**Abstract:** In this paper, aiming at the problem of poor quality and low spraying efficiency of irregular for complex freeform surfaces, a new spray painting robot trajectory planning method based on point cloud slicing technology is proposed. Firstly, the point cloud data of the workpiece to be sprayed is obtained by laser scanning. The point cloud data is processed to obtain the point cloud model of the sprayed workpiece. Then the section polysemy line is obtained after slice acquisition and section data processing of the point cloud model. The section polysemy line is sampled on average, and the normal vector of all sampling points is estimated. Finally, interpolation algorithm is used to connect the data points to obtain the space trajectory of spraying robot. In addition, the droplet trajectory model for electrostatic spray painting is established. The experimental results show that the method fully meets the requirements of coating thickness and improves the spraying efficiency and uniformity of coating.

**Keywords:** point cloud slicing; surface modeling; trajectory planning; electrostatic spray painting

## 1. Introduction

Automatic painting is an important process in the manufacture of many products, such as automobiles, furniture, and airplanes [1]. With the development of various science and technology, spraying robots have been widely used, such as in agricultural production [2], pest control [3], art creation [4,5], art restoration [6], etc. People have higher requirements for the spraying quality of the product surface, but the original spraying robot spraying model and the trajectory optimization theory have been unable to adapt to these higher and higher production requirements. To achieve a new spray painting standard and to realize the production target with high efficiency and low cost, research on the new spray modeling and the optimization algorithm and control strategy of the spraying robot with high performance have become focused issues both at home and abroad.

The surface modeling of spraying workpiece is the key of spraying trajectory planning. Because of the diversity and complexity of spraying workpiece, it is not easy to find a complete set of practical surface modeling methods for spraying workpieces [7–9]. At present, the modeling method based on parametric surface and the surface modeling method based on CAD model are used in spraying robots [10]. The modeling method based on parametric surface is based on approximation, and polygons are used to generate curves or surfaces, so as to complete the surface modeling of the workpiece. The parametric surface expression obtained by this method has high precision, but the expression form is very complex, which is difficult to be applied to the surface modeling of complex workpiece with large curvature [11]. The surface modeling method based on a CAD model refers to the fact that the CAD model data of the workpiece was obtained before the surface modeling, and the triangular

mesh surface is used to approximate the surface of the spraying workpiece, so as to plan the spraying trajectory of the spraying robot [12]. Alessandro et al. proposed a practical CAD model-based trajectory optimization method for spraying robot [13]. However, how to carry out trajectory planning on free surface without CAD data model is a difficult problem. Therefore, the current surface modeling method is not suitable for surface modeling and trajectory planning of spraying workpiece with large curvature and no CAD model data, which seriously affects the spraying efficiency and effectiveness. After the surface modeling of painting workpiece is completed, the next step is to plan the space trajectory of painting robot, and finding the appropriate space trajectory of painting robot is very important to improve the spraying efficiency and quality [14]. In recent years, scholars have made a lot of achievements in the research of spraying robots [15–20]. Professor Chen Heping's team from the United States optimized the key parameters, thus improving the trajectory optimization method [21]. Pal et al. proposed a multi-objective optimization algorithm with spraying time and uniformity of coating thickness as the optimization objectives, but the trajectory optimization method is too complex and has poor practicability [22]. In reference [23], a method is presented for lowering the energy consumption and/or increasing the speed of a standard manipulator spray painting a surface. Anurag et al. applied spraying robots to different industries and proposed a new trajectory optimization method adapted to this application field [24]. To sum up, most of the current international studies focus on trajectory optimization methods, and there are few studies on the generation of complex surface spray trajectory without CAD data model or large curvature change. Moreover, trajectory optimization methods are too complex and less practical. To obtain better spraying effect, it is necessary to study the surface modeling method of spraying workpiece and the space trajectory planning method of spraying robot. The development of bionic robot technology provides inspiration for the control scheme of spraying robot. For example, apply neuroscience to spray robots, using goal-directed neural circuits to improve spray robots [25]. Similarly, the study of the compensation relationship between biological cognition and movement can enhance the cooperative relationship between the joints of the spraying mechanical arm, so as to achieve accurate control of the spray gun position and improve the spraying efficiency [26,27]. Living organisms are far superior to state-of-the-art robots as they have evolved a wide number of capabilities that far encompass our most advanced technologies [28]. We can get inspiration from the biological principle and combine the biological model with spraying robot control technology. This is a complex process, but it provides a new direction for the improvement and development of spraying robot.

The surface modeling method based on point cloud patchation technology is a surface modeling method based on workpiece scanning system, which is suitable for small batch and many kinds of workpiece surface spraying. The innovation of spraying trajectory planning based on point cloud slicing technology lies in the use of point cloud data as the data format of surface modeling and trajectory generation of spraying robot. The idea of trajectory planning for spray painting robot based on point cloud slicing technique is: first, the point cloud data is obtained by laser scanning the workpiece to be sprayed, and then the point cloud data is reconstructed in three dimensions to complete the surface modeling of the sprayed workpiece. According to the point cloud slicing technique mentioned, the cross-section data of patches can be simplified, smoothed, and segmented, and the effective data points of the workpiece surface can be obtained. We can get the projection point of the end effector on the workpiece surface after these data points are average sampled. These offset data points contain the position and direction information of the end effector in the spraying process. Through the bias algorithm, the offset of the sampling point in the normal direction can be calculated. Finally, the interpolation algorithm is used to connect the logarithmic points to form a continuous spatial trajectory of the robot. When there are more kinds of spraying workpiece or the curvature of spraying workpiece changes greatly, the point cloud technology is suitable for the surface modeling and trajectory planning of the spraying robot. This technique not only reduces the complexity of the spraying process but also controls the position of the spraying gun, thus improving the quality and efficiency of spraying.

## 2. Spray Workpiece Surface Modeling

The surface modeling method of spraying workpiece based on point cloud slicing technology is a surface modeling method based on the workpiece scanning system, which can be applied to the surface modeling of spraying workpiece with large curvature changes. By slicing the pre-processed point cloud with the help of data acquisition and data processing technology in reverse engineering, the section profile can be extracted as the initial spraying path of the spraying robot.

### 2.1. Data Acquisition and Processing of Point Cloud Slice of Spraying Workpiece

To reduce the modeling time of complex surface and improve the spraying efficiency, the point cloud data on the surface of complex spraying workpiece can be obtained through the scanning system, and the section data can be directly generated from the point cloud data, and then the section data can be processed. As shown in Figure 1, The steps are as follows: (1) determine the number of slice layers; (2) Separate and calculate the slice data; (3) Polysemy line reconstruction. The specific section modeling process of three-dimensional entity can be referred to the reference [29].

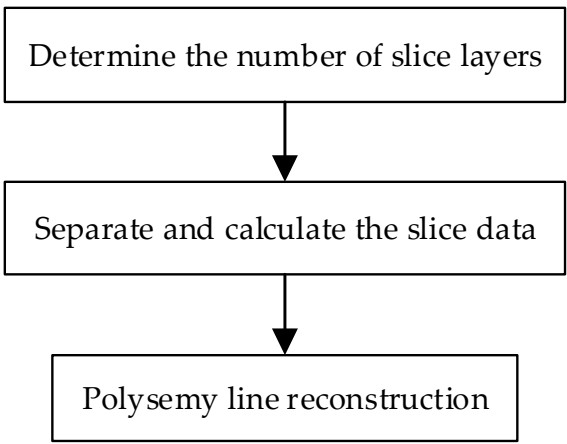

**Figure 1.** Point cloud slicing process.

The section data obtained after slicing the spraying workpiece usually have problems such as redundancy, noise, and connection errors, which makes it difficult to meet the needs of subsequent spraying trajectory planning. Therefore, it is necessary to process the section data. This process has two main purposes: First, reduce the error of cross-section data and improve the accuracy of trajectory planning. The second is to facilitate the subsequent matching and pose generation, reduce the trajectory generation time, and improve the spraying efficiency. Cross-section data processing mainly includes three steps: point cloud data streamlining, discrete curvature estimation and section data segmentation based on feature points.

### 2.1.1. Data Streamlining

Point cloud data set of surface shape characteristics of spraying workpiece can be expressed by mathematical expression as: $\Omega = \{P_i(x_i, y_i, z_i) i = 1, 2 \cdots n\}$. In the expression, $\Omega$ represents a point cloud, $P_i(x_i, y_i, z_i)$ represents an arbitrary point in the point cloud, $n$ represents the total number of point clouds. We can use the angle method to streamline the cloud data. Figure 2 is a schematic illustration of the angle method. The specific steps are as follows:

(a)　Set the angle threshold as $[a_{min}, a_{max}]$. Calculate the corresponding cosine value. As $\alpha_{min}$ and $\alpha_{max}$ are generally obtuse, so $0 > cos(\alpha_{min}) > cos(\alpha_{max})$.

(b) Calculate the cosine of the angle *cosα* of the adjacent three points $P_{i-1}, P_i, P_{i+1}$ in the cross section data:

$$\cos\alpha = \frac{\|P_{i-1}P_i\|^2 + \|P_iP_{i+1}\|^2 - \|P_{i-1}P_{i+1}\|^2}{2\|P_{i-1}P_i\|\|P_iP_{i+1}\|} \tag{1}$$

If $P_{i-1}P_i$ or $P_iP_{i+1}$ is 0, then let $cos\alpha = cos(\alpha_{max})$ in order to solve the key issues existing in the patch data. If the cross-section polyline is closed, it is necessary to calculate the cosine of the angle at the starting point $P_0$ (or terminal point $P_n$). Then we have $i = (1, \ldots, n-1)$. When $i = 0$, let $P_{i-1} = P_{n-1}$ in order to solve the problem. If the cross-section polyline is not closed, there is no angle at the starting point and the terminal point. Then we have $i = (1, \ldots, n-1)$.

(c) If $cos\alpha \in [cos(\alpha_{max}), cos(\alpha_{min})]$, then delete the center point $P_i$, that is $P_i = P_{i+1}, \ldots, P_{n-1} = P_n$, $n = n - 1$. If $cos\alpha \notin [cos(\alpha_{max}), cos(\alpha_{min})]$, let $i = i + 1$.

(d) If *i* satisfies the termination condition, end the loop. Otherwise, turn to (b) to continue calculation.

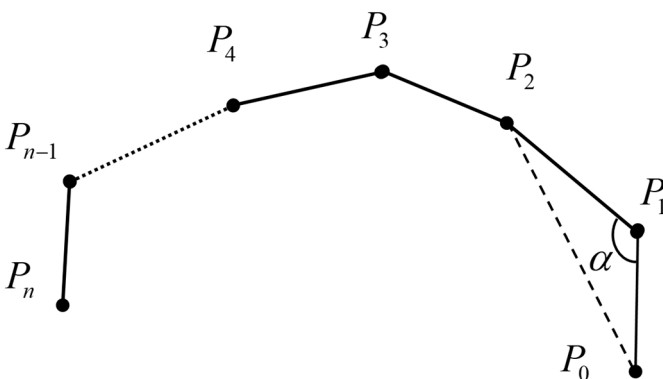

**Figure 2.** Diagram of angle method.

### 2.1.2. Discrete Curvature Estimation

Discrete curvature estimation is mainly used for denoising, feature extraction and data segmentation for cross-section data. The circular curvature method is used to obtain the discrete curvature of the cross-section data. To facilitate the description, the curvature of the cross-section data is divided into center point and endpoint two cases to discuss.

(a) Curvature estimation of the center point

The curvature $C_i$ of a point $P_i$ among the middle point array $P_1, \ldots, P_{n-1}$ is determined by the circle obtained by the three adjacent points $P_{i-1}, P_i, P_{i+1}$, As shown in Figure 3, the formula is as follows:

$$C_i = \frac{1}{R_i} = \frac{4S_{\Delta P_{i-1}P_iP_{i+1}}}{\|P_{i-1}P_i\|\|P_iP_{i+1}\|\|P_{i-1}P_{i+1}\|} \tag{2}$$

where $S_{\Delta P_{i-1}P_iP_{i+1}}$ is the area of a signed triangle. When $P_{i-1}, P_i, P_{i+1}$ is counterclockwise, the sign is positive. On the contrary, the sign is negative. The sign can be determined by the vector fork. The area of signed triangular can be calculated as follows:

$$S = \sqrt{s(s - \|P_{i-1}P_i\|)(s - \|P_iP_{i+1}\|)(s - \|P_{i-1}P_{i+1}\|)} \tag{3}$$

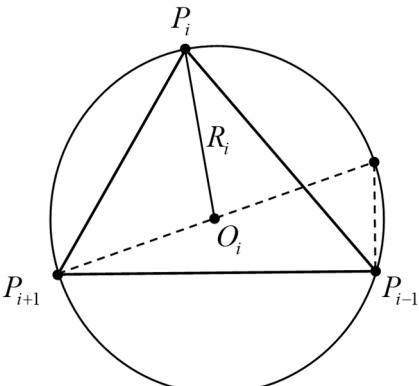

**Figure 3.** Curvature estimation of the center point.

(b) The curvature estimation of endpoints $P_0$ and $P_n$

If the cross-section polyline is closed, then the endpoints $P_0$ and $P_n$ are the same point. When $i = 0$, let $P_{i-1} = P_{n-1}$ so that the curvature of $P_0$ can be obtained with the Equation (3). If the cross-section polyline is not closed, the calculation method for the curvatures of endpoints $P_0$ and $P_n$ is: Add a point $P_{0-}$ and $P_{n+}$ in the opposite direction of the cut vectors of two endpoints. The curvatures of $P_0$ and $P_n$ can be calculated by $P_{0-}, P_0, P_1$ and $P_{n-1}, P_n, P_{n+}$ respectively. Figure 4 shows the curvature estimation for endpoint $P_0$, and the unit tangent vector $\vec{t_0}$ is:

$$\vec{t_0} = \frac{\vec{r_0} \times \vec{v_0}}{\|\vec{r_0} \times \vec{v_0}\|} \tag{4}$$

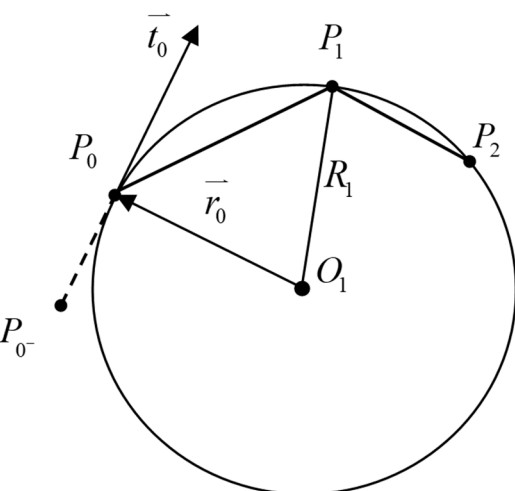

**Figure 4.** Curvature estimation of endpoint $P_0$.

Here, $\vec{v_0} = \vec{P_2P_0} \times \vec{P_1P_0}$, $\vec{r_0} = \vec{O_1P_0}$. Suppose the center coordinate of circle $O_1$ is $(x_c, y_c)$, then we have:

$$\begin{cases} (x_c - x_0)^2 + (y_c - y_0)^2 = R_1^2 \\ (x_c - x_1)^2 + (y_c - y_1)^2 = R_1^2 \\ (x_c - x_2)^2 + (y_c - y_2)^2 = R_1^2 \end{cases} \tag{5}$$

Solve the equation set above, then we can get:

$$\begin{cases} x_c = \frac{a-b+c}{d} \\ y_c = \frac{f-e-g}{d} \end{cases} \tag{6}$$

where $a = (x0 + x1)(x1 - x0)(y2 - y1)$, $b = (x2 + x1)(x2 - x1)(y1 - y0)$, $c = (y0 - y2)(y1 - y0)(y2 - y1)$, $d = 2[(x1 - x0)(y2 - y1) - (x2 - x1)(y1 - y0)]$, $e = (y1 + y0)(y1 - y0)(x2 - x1)$, $f = (y2 + y1)(y2 - y1)(x1 - x0)$, $g = (x0 - x2)(x1 - x0)(x2 - x1)$.

Add a point $P_{0-}$ in the opposite direction of $\overrightarrow{t_0}$ and make it satisfy $\|P_{0-}P_0\| < \|P_0P_1\|$. The Equation (2) can be used to solve the curvature after three adjacent points $P_{0-}, P_0, P_1$ are obtained. Similarly, the curvature of $P_n$ can be calculated by adding point $P_{n+}$.

### 2.1.3. Cross-section Data Segmentation Based on Feature Points

The cross-section characteristic curve is generally the point whose position or tangential vector is continuous. Therefore, the extraction of the feature points on the cross-section curve is dominated by these two types of points. Find the feature points in the section data, then the curve can be divided into multiple segments by referring to these points. The first order difference can be used to calculate the derivative of the discrete curve $C$ at point $P_j$:

$$C'(P_j-) = (P_{j-1} - P_{j-2}) + (P_j - P_{j-2})\frac{\|P_j - P_{j-1}\| + \|P_{j-1} - P_{j-2}\|/2}{\|P_j - P_{j-1}\|/2 + \|P_{j-1} - P_{j-2}\|/2} \tag{7}$$

$$C'(P_j+) = (P_{j+2} - P_{j+1}) + (P_{j+2} - P_j)\frac{\|P_{j+1} - P_j\| + \|P_{j+2} - P_{j+1}\|/2}{\|P_{j+1} - P_j\|/2 + \|P_{j+2} - P_{j+1}\|/2} \tag{8}$$

If the expressions above satisfies $\left|C'_i(P_j-) - C'_i(P_j+)\right| > L$ and $\theta = angle\left(C'_i(P_j-), C'_i(P_j+)\right) > \alpha$, then the data point $P_j$ is a sharp point, where $L$ and $\alpha$ are the set length and angle threshold respectively. Similarly, if they satisfy $\left\|\|V''_i(P_j-)\| - \|V''_i(P_j+)\|\right\| \geq N$ ($N$ is the set threshold), $P_j$ is considered to be the first order continuous point on $C$.

### 2.2. Experiment on Surface Modeling of Spraying Workpiece

With VC++ 6.0 as the development platform, the trajectory optimization software system based on point cloud slicing technology is designed. The main functions of the surface modeling subsystem in the system include input and output of point cloud data, data acquisition of point cloud patches, and cross-section data processing.

The main functions of the surface modeling subsystem of spraying workpiece are described as follows:

1. Input and output of point cloud data. For *.txt file format, we use C++ input and output stream function to achieve read and storage of arbitrary point cloud data.
2. Data acquisition of point cloud patches. Include patch thickness calculation, patching direction definition, plane and point cloud intersection, polyline construction and other functions, which can achieve the slicing of the point cloud model. Polyline construction includes patch data sorting and polyline orientation adjustment.
3. Cross-section data preprocessing. Include cross-section data streamlining, discrete curvature estimation, cross-section data segmentation and other functions.

The spraying workpiece is shown in Figure 5. The ATOSII three-dimensional optical scanner manufactured by GOM company in German is used as measurement equipment, Point Cloud Model of the Workpiece as shown in Figure 6. The workpiece surface is labeled with black background and

white dots. When a scanner scans an artifact in blocks, scan at least three labels at a time. Point cloud data preprocessing is implemented in Imageware. It mainly includes the following four steps:

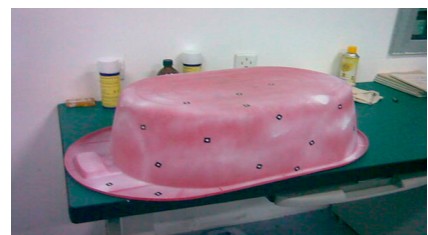

**Figure 5.** Spraying workpiece.

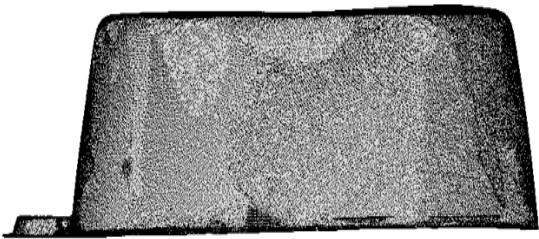

**Figure 6.** Point cloud model of the workpiece.

### 2.2.1. Delete Outlier Data

According to the distribution of abnormal points, we can use different commands to delete them. When the distribution is more intensive, we can use Modify → Extract → Circle-Select Points to circle and delete. When the distribution is more sparse, we can use Modify → Scan Line → Pick Delete Points to delete by point. We can also use Create → Points to generate a new point cloud at the abnormal point, then use Modify → Extract → Subtract Cloud from Cloud to subtract the outlier data.

### 2.2.2. Data Interpolation

There will be lost points which need to be interpolated in the groove, hole, and the occluded area of the surface. For scanning line point cloud or section data, we can use Modify → Scan Line → Fill Gap to perform linear or non-linear interpolation. For scattered cloud data, we can use Construct → Cross Section → Cloud Parallel to generate cross-section data. Then we can use Modify → Scan Line → Fill Gap to fill the data. We can also use Construct → Points → Sample Curve or Construct → Points → Sample Surface to discrete fitting curves or surfaces so as to generate missing points.

### 2.2.3. Data Smooth

To reduce or eliminate noise, we can use Modify → Smooth → Scan to perform smoothing process. If the point cloud has changes in corners, we can also use Modify → Smooth → Scan Between Corners to smooth the data at the transition.

### 2.2.4. Data Streamlining

Uniform sampling method, string error method and spatial sampling method can be select to perform data streamlining. The corresponding command is Modify → Data Reduction → Sample Uniform, Modify → Data Reduction → Chordal Error and Modify → Data Reduction → Space Sampling. Among them, the space sampling method can delete the data by setting a distance tolerance, which has a better streamlining effect on the overlapping area of the point cloud. The streamlining effect is also the best. To ensure the quality of follow-up point cloud patches, we usually take space sampling method (distance tolerance is 0.4 mm) for data streamlining. While preserving the shape features, the data of the overlapping area is effectively deleted.

The shape of point cloud data after pre-processing is shown in Figure 7. After the pre-processing of point cloud data, point cloud slice acquisition and section data processing are carried out. Taking the z-axis as an example, the slice direction was selected as the z-axis and the slice layer number is 20. The slices of the resulting plastic basin are shown in Figure 8. Because the curvature of the workpiece changes greatly, there are some blind areas in the directions of X, Y and Z slices, but the three slices are complementary.

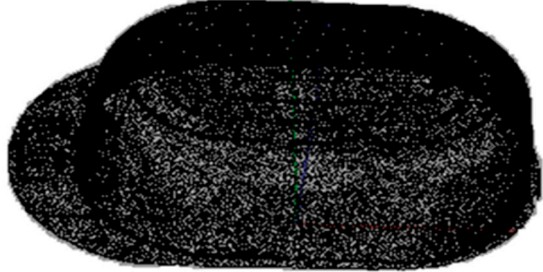

**Figure 7.** Point cloud data preprocessing modeling.

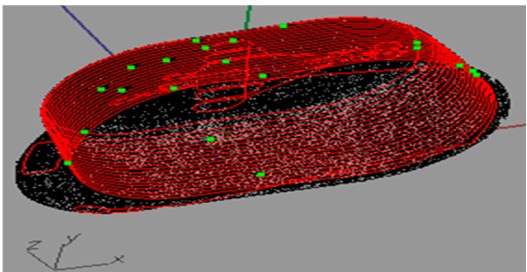

**Figure 8.** Point cloud slice along the Z-axis.

From the experimental results, since most of the painting paths are parallel to each other, the patch can be arranged in a parallel plane which is uniformly distributed along the patching direction. The patching direction is perpendicular to the painting path, which determines the direction of the patch curve. For the same sprayed area, the position and number of patches will change along different directions, so that different sections are obtained. The direction of the generated painting path based on the contour lines of these sections is also different. The number of layers should be determined by factors such as the curvature of the workpiece surface, the diameter of spray painting and the size of the workpiece. For the workpieces with large curvature changes, the minimum values of all patch thickness can be obtained by adaptive patching techniques. The minimum patch thickness can be used as the patch thickness for uniform slice.

## 3. Spatial Trajectory Planning for Spray Painting Robot Based on Point Cloud Slicing Technique

According to the point cloud slicing technique mentioned, the cross-section data of patches can be simplified, smoothed, and segmented, and the effective data points of the workpiece surface can be obtained. We can get the projection point of the end effector on the workpiece surface after these data points are average sampled. Since the position and orientation of the end effector includes both position and attitude parameters, and the slicing data only includes position information, so these data points cannot be used as the position and orientation of the end effector directly. During the operation of the spray painting robot, it is necessary to keep the axis of the end effector perpendicular to the workpiece surface and the distance from the workpiece surface constant. To ensure the uniform painting distribution, a corresponding algorithm has been designed to calculate the offset of sampling point in the normal line direction. These offset data points contain information about the position and

orientation of the end effector during the spray painting process. Finally, the interpolating algorithm is used to connect the data points in order to form a continuous spatial path of the robot.

### 3.1. Mathematical Model of the Position and Orientation of End Effector

Suppose the workpiece is stationary in the Cartesian coordinate system XYZ, we can directly get the function expression of the surface as z = h (x, y). Here, the projection is h: $D \rightarrow R$ (real number field), the domain is $D \subset R^2$. By the general concept of the set, function expression of the surface can be defined as:

$$S = \left\{ (x, y, z) \big| z = h(x, y), (x, y) \in D \right\} \tag{9}$$

Assuming that the position of the end effector Tool Center Point (TCP) relative to the fixed Cartesian coordinate system XYZ is represented by a three-dimensional vector function *P(t)*. The orientation of the end effector relative to the fixed Cartesian coordinate system XYZ is represented by another three-dimensional vector function *O(t)*. The relationship between these two vectors and time can be expressed as:

$$P(t) = \left[ P_x(t)\ P_y(t)\ P_z(t) \right]^T \tag{10}$$

$$O(t) = \left[ O_x(t)\ O_y(t)\ O_z(t) \right]^T \tag{11}$$

The vectors $P_x(t)$, $P_y(t)$, $P_z(t)$ represent the position of the end effector in the XYZ coordinate system at time t. The vectors $O_x(t)$, $O_y(t)$, $O_z(t)$ represent the rotation angles of the end effectors around the X, Y and Z axis at time t. This rotation system relative to a fixed coordinate system is generally referred to as a 'rolling, pitching, yawing' system. It is easy to define a rotation matrix based on these angle values, so that the fixed coordinate system can be converted into the rotating coordinate system of the end effector by premultiplying or postmultiplying the matrix. For convenience, a vector function *a(t)* is defined to generally represent the position and orientation of the end effector.

$$a(t) = \left[ P(t)\ O(t) \right]^T \tag{12}$$

### 3.2. Spatial Trajectory Acquisition of Spray Painting Robots

The spatial trajectory acquisition of spray painting robots can be divided into three steps: setting the number of sampling points, estimating the normal vector of sampling points, and generating spatial trajectories.

#### 3.2.1. Setting the Number of Sampling Points

We can obtain the projection of the painting trajectory on the workpiece surface after the average sampling of the patches which have already been processed. The number of sampling points (recorded as *num*) is set by the artificial interaction method. Equation (13) is used to calculate the interval distance dm of the sampling points on each patch, and the coordinate data of the trajectory points can be obtained.

$$d_m = \frac{\sum\limits_{i=0}^{N-1} \|P_i P_{i+1}\|}{num} \tag{13}$$

where *N* denotes the number of points on the polyline of the patch, $P_i$ and $P_{i+1}$ denote the *i*-th and (*i*+1)-th sampling points on the polyline.

#### 3.2.2. Estimating the Normal Vector of Sampling Points

According to the start point for operation and the moving direction of the end effector on the first painting trajectory, the sorting direction of patch data on the first layer should be determined first. In addition, the sorting direction of patch data on adjacent layer is set in reverse order, so that the

painting trajectory of the end effector on the workpiece surface can be determined. Painting trajectory is a set of data points formed by a set of patches, known as the sampling point set $Q$:

$$Q = \left\{ Q_i(x_j, y_i, z_i), i = 1, 2, 3 \dots, t \right\} \tag{14}$$

where $x_j$ is the X-coordinate of the $j$-th patch, $i$ is the number of sampling points on the patch of the $i$-th layer, $y_i$, $z_i$ are the Y-coordinate and Z-coordinate of the sampling point on the $i$-th layer. There are two main methods for estimating the normal vector of each data point $Q_i$ in the sampling point set of the microcutting plane: One is to make use of the neighborhood information of the points. By computing the $k$-neighborhood of the points, the best approximation plane of the $k$ neighboring points can be constructed by the least squares method. The plane can be regarded as the microcutting plane at this point. The normal vector of the plane is the normal vector of the desired sampling point [30–32]. The microcutting plane in this method has ambiguity. To make the plane of the normal vector point to the outside of the surface, we need to adjust the direction of the normal vector. Another method is to use the triangular topology of scattered data, which can be obtained by the normal vector of the relevant triangles of the points. Then the direction of the normal vector can be determined by the direction of the directed triangles directly [33,34]. Here, we use this method to estimate the normal vector of sampling point.

As is shown in Figure 9, there are $m$ point $P_j$ ($j = 1, 2, \dots, m$) around the sampling point $Q_i$, which are adjacent to each other. Thus, m related triangles are formed by these points. The unit normal vector of the triangle formed by $Q_i$, $P_j$, $P_{j+1}$ is $\vec{n}_j$:

$$\vec{n}_j = \frac{(P_j - Q_j) \times (P_{j+1} - Q_j)}{\|(P_j - Q_j) \times (P_{j+1} - Q_j)\|} \tag{15}$$

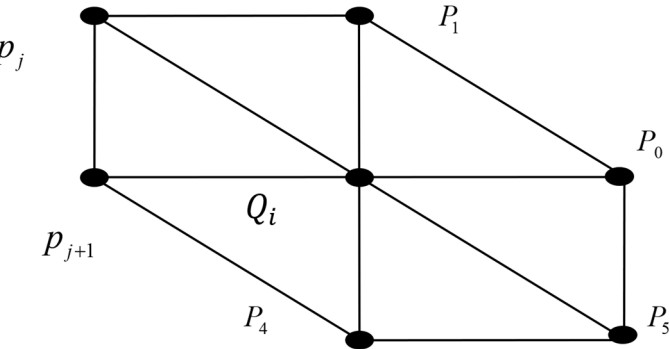

**Figure 9.** Estimation of normal vector of point $Qi$.

In this expression, ($j = 1, 2, \dots, m$, and $P_{m+1} = P_1$). If the area of the triangle is $S_j$, by weighting combination and using $\sigma_j = 1/s_j$ as the weighting factor, the normal vector $\vec{n}$ of point $Q_i$ is estimated as:

$$\vec{n} = \sum_{j=1}^{m} \sigma_j \vec{n}_j \tag{16}$$

### 3.2.3. Generating Spatial Trajectory

According to the requirements of the spray painting robot, the perpendicular distance from the end effector to the sprayed workpiece surface is set as the parameter Offset Distance, which is denoted as $H$, while the position and orientation of the end effector can be obtained by the algorithm above.

The bias point $Q_i$ of point $Q_i$ can be obtained by setting offset distance $H$ of point $Q_i$ in the direction of normal vector $\vec{n}$. The mathematical expression is as follows:

$$O_i = Q_i + H\vec{n} \tag{17}$$

Point $O_i$ contains the coordinate values and the unit normal vectors, which gives the position (coordinate) and direction (opposite to the direction $\vec{n}$) of the end effector. We can get a set of offset points $O$ by using the same method to traverse all the points in the sampling point set $Q$. The information contained in the entire point set $O$ represents the position and orientation parameters of the end effector during the spray painting process. Finally, the points are connected to form a continuous trajectory of the spray painting robot by using the spatial line or arc interpolation algorithm.

For the point cloud slice shown in Figure 8, the number of data points to be extracted is 20 and the offset distance is 0. The resulting painting trajectory is shown in Figure 10.

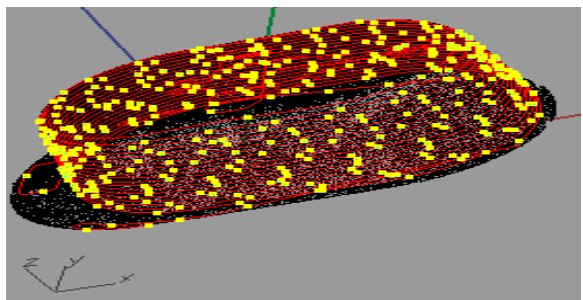

**Figure 10.** Spray trajectory of point cloud patch in Z-axis direction.

## 4. Droplet Trajectory Model for Electrostatic Spray Painting

In modern automobile spraying, in order to improve the use rate of coating, electrostatic spraying method is adopted. There are many electrostatic spraying parameters, compared with air spraying, robot trajectory planning is more difficult [35–37]. The following is a practical electrostatic spraying trajectory planning method based on the above spraying trajectory planning method. The Lagrangian method is used to simulate the droplet movement trajectory of electrostatic spray painting. Here, we use the improved formation of the Basset-Boussinesq-Oseen (BBO) equation of the well-known particle motion equation in fluid mechanics to describe the trajectory of the electrostatic droplet [38]. The improvement is mainly reflected in the BBO equation by taking the influence of the electrostatic field into account. The BBO equation is a form of Newton's second law of motion that contains all the forces acting on the particles, which are steady-state air thrusts $F_D$, buoyancy, virtual inertial forces, Basset forces, gravity $F_g$ and electrostatic force $F_{E/S}$. The improved BBO equation is as follows:

$$m\frac{dv}{dt} = 3\pi\mu D(u - v) + \frac{\pi}{6}D_g^3\rho_g g\frac{du}{dt} + \frac{\pi}{12}D^3\rho\frac{d(u-v)}{dt} + \frac{\frac{3}{2}D^2\sqrt{\pi\rho_g\mu}[\int_0^t \frac{u-v}{\sqrt{t-t'}}dt' + \frac{(u-v)_0}{\sqrt{t}}]}{4} + \frac{mg}{5} + \frac{qE}{6} \tag{18}$$

In the equation above, $u, D_g, \rho_g, q$ are the droplet instantaneous velocity, droplet diameter, droplet density, droplet charge respectively. $v, D, \rho, t$ are the particle instantaneous velocity, particle diameter, particle density, particle movement time respectively. $\mu$ is the viscosity of the paint. $E$ is the electrostatic potential field. The simplified model of the electrostatic droplet trajectory can be expressed as:

$$m\frac{dv}{dt} = \sum F = F_D + F_{E/S} + F_g \tag{19}$$

Only two major forces will affect the momentum of the droplet, i.e., the thrust of the surrounding turbulent air and the electrostatic force of the charged droplet in the electrostatic field. In addition, turbulent air thrust can be introduced by Stokes law:

$$\boldsymbol{F}_D = 3\pi\mu_g D_p f(\boldsymbol{u} - \boldsymbol{v}) \tag{20}$$

$\mu_g, D_p, f, \boldsymbol{u} - \boldsymbol{v}$ are the air viscosity, droplet diameter, resistance factor and the relative velocity of ambient air $u$ and droplet $v$ respectively. The resistance factor is defined as follows:

$$
\begin{aligned}
f &= 1 + \frac{\mathrm{Re}_r^{2/3}}{6}, \mathrm{Re}_r < 1000 \\
f &= 0.0183\,\mathrm{Re}_r, 1000 \le \mathrm{Re}_r < 3 \times 10^5
\end{aligned} \tag{21}
$$

$$\mathrm{Re}_r \equiv \frac{D_p|\boldsymbol{u} - \boldsymbol{v}|\rho}{\mu_g} \tag{22}$$

Equations (21) and (22) are based on the instantaneous velocity of the gas, and the instantaneous velocity is shown as (23):

$$\boldsymbol{u} = \overline{\boldsymbol{u}} + \widetilde{\boldsymbol{u}} \tag{23}$$

$\overline{\boldsymbol{u}}, \widetilde{\boldsymbol{u}}$ are the average gas velocity and the turbulent velocity respectively.

The electrostatic force is the product of the droplet charge $q$ and the electrostatic field $E$, which is the divergence of the electrostatic potential. Therefore, the droplet trajectory equation can be written as:

$$m\frac{d\boldsymbol{v}}{dt} = m\frac{d^2\boldsymbol{x}}{dt^2} = 3\pi\mu_g D_p f(\boldsymbol{u} - \boldsymbol{v}) - q\nabla\Phi \tag{24}$$

Equation (24) is a vector equation, which can be solved by the fixed-step $\Delta t$ Euler time integral.

## 5. Spraying Experiment

The plastic basin is replaced with a certain brand of car body as the spraying experiment object. Firstly, the three-dimensional optical scanner is used to measure the point cloud data of the experimental vehicle, and then the workpiece point cloud model is obtained by using the automatic point cloud assembly technology of the scanner. The pre-processing of automobile body point cloud data is realized in reverse engineering software Imageware. The car body shape obtained after pre-processing of point cloud data is shown in the prototype system as shown in Figure 11.

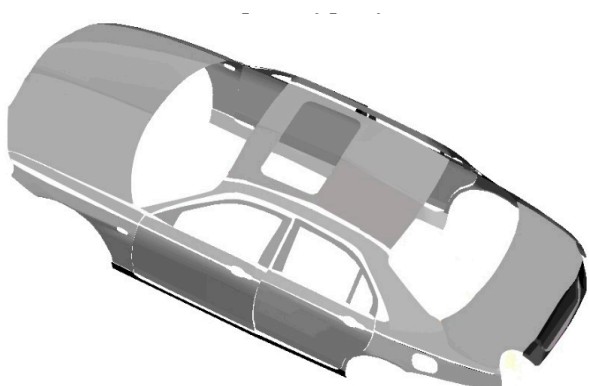

**Figure 11.** Car body shape after pre-processing of point cloud data.

We use ABB electrostatic painting robot to conduct painting experiments on the body, as shown in Figure 12. Multiple ABB spraying robots are used to spray the body in different areas on the same automobile spraying line, as shown in Figure 13. Spraying experiment adopts the droplet trajectory model of electrostatic spraying, and combined with the installation location distribution of ABB robot,

the body is divided into three parts: the roof and rear, the left side of the body, and the right side of the body. Trajectory planning is carried out on the three parts by using the trajectory planning method of spraying robot based on point cloud slicing technology. The point cloud model is patched by setting the slicing direction (relative to painting path direction) and patch number (relative to the number of reciprocating strokes). After the polyline of patch is obtained, the average sampling is performed. Then the normal vectors of all sampling points are estimated. Finally, the spatial path of the spray painting robot is obtained by the interpolation algorithm. In the process of trajectory planning, according to the curvature of different body surfaces, different trajectory spacing is adopted according to the characteristics of the experimental platform. The painting effect is better when the robot end moves at a uniform speed.

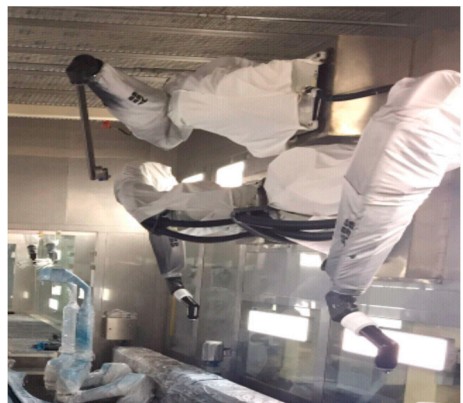

**Figure 12.** ABB spraying robot.

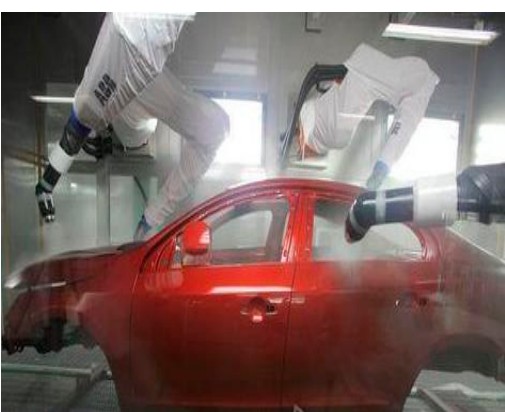

**Figure 13.** Automobile spraying line.

The spray tracks generated on the right side of the left side of the car roof are shown in Figures 14–16 respectively. Among them, the roof and rear of the vehicle are divided into three tracks, the left part of the vehicle is divided into eight tracks, and the right part of the vehicle is divided into eight tracks.

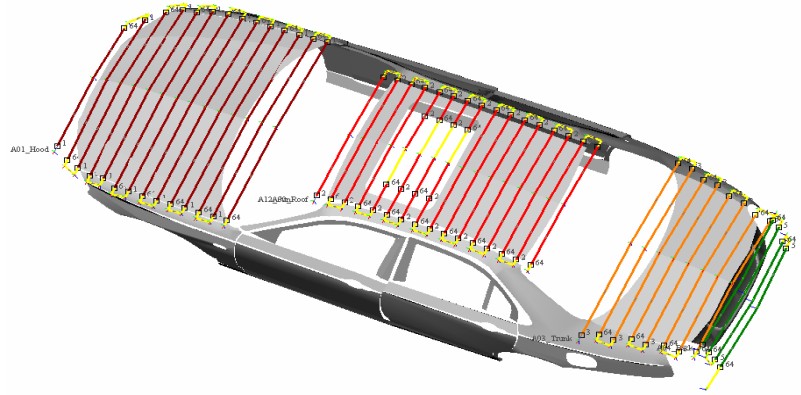

**Figure 14.** Spray Painting trajectory on the roof and rear part of the automobile.

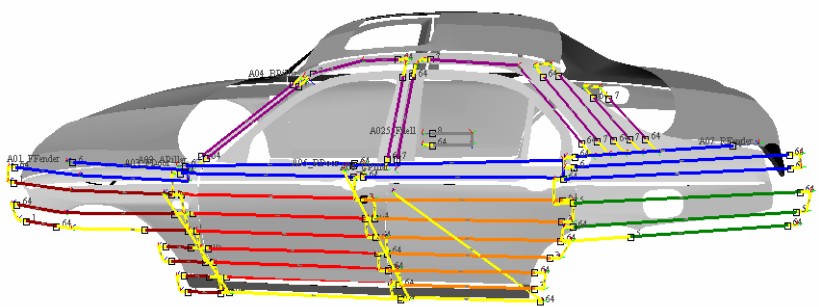

**Figure 15.** Spray Painting trajectory on the left part of the automobile.

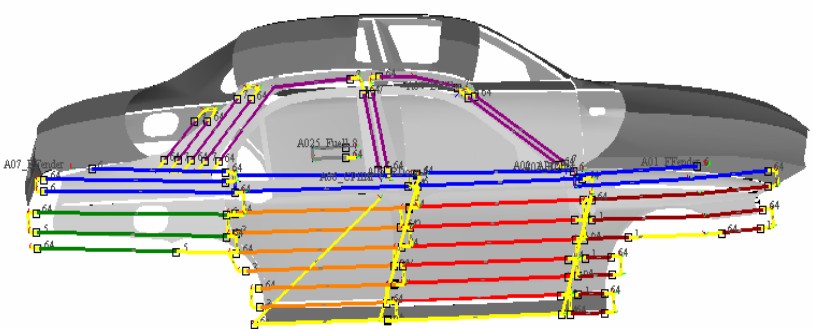

**Figure 16.** Spray Painting trajectory on the right part of the automobile.

The tracks of different segments are displayed in different colors in the off-line programming system of the spraying robot. The spray parameters corresponding to the track of each part of the body are shown in Tables 1–3, respectively. The parameters Del, AM, SA1, SA2, and HV represent the paint flow rate, rotating speed of rotary bell, flow of air 1, flow of air 2, and electrostatic voltage of rotary bell respectively. Air 1 is used to drive the turbine and drive the rotary bell to rotate, air 2 is the shaping air. The paint flow of the yellow track is 0, and there is no spraying at this time, just the position transfer of ABB spraying robot spraying.

**Table 1.** Operating parameters of the spray painting trajectory on the roof and tail.

| Trajectory | | Operating Parameters of Spray Trajectory on the Roof and Tail Parts of the Tested Automobile | | | | |
|---|---|---|---|---|---|---|
| NO. | Color | Del (L/min) | AM (KRPM) | SA1 (L/min) | SA2 (L/min) | HV (KV) |
| 1 | ▬▬▬ | 290 | 40 | 250 | 180 | 60 |
| 2 | ▬▬▬ | 240 | 40 | 250 | 180 | 60 |
| 3 | ▬▬▬ | 300 | 40 | 250 | 220 | 60 |
| 4 | ▬▬▬ | 200 | 40 | 250 | 180 | 60 |
| 6 | ▬▬▬ | 280 | 40 | 250 | 180 | 60 |
| 7 | ▬▬▬ | 0 | 40 | 280 | 180 | 60 |

**Table 2.** Operating parameters of the spray painting trajectory on the left.

| Trajectory | | Operating Parameters of Spray Trajectory on the Left of the Tested Automobile | | | | |
|---|---|---|---|---|---|---|
| NO. | Color | Del (L/min) | AM (KRPM) | SA1 (L/min) | SA2 (L/min) | HV (KV) |
| 1 | ▬▬▬ | 340 | 40 | 250 | 180 | 60 |
| 2 | ▬▬▬ | 100 | 40 | 250 | 180 | 60 |
| 3 | ▬▬▬ | 310 | 40 | 250 | 180 | 60 |
| 4 | ▬▬▬ | 230 | 40 | 250 | 180 | 60 |
| 5 | ▬▬▬ | 315 | 40 | 250 | 180 | 60 |
| 6 | ▬▬▬ | 285 | 40 | 250 | 180 | 60 |
| 7 | ▬▬▬ | 285 | 40 | 250 | 180 | 60 |
| 8 | ▬▬▬ | 0 | 40 | 280 | 180 | 60 |

**Table 3.** Operating parameters of the spray painting trajectory on the right.

| Trajectory | | Operating Parameters of Spray Trajectory on the Right of the Tested Automobile | | | | |
|---|---|---|---|---|---|---|
| NO. | Color | Del (L/min) | AM (KRPM) | SA1 (L/min) | SA2 (L/min) | HV (KV) |
| 1 | ▬▬▬ | 340 | 40 | 250 | 180 | 60 |
| 2 | ▬▬▬ | 100 | 40 | 250 | 180 | 60 |
| 3 | ▬▬▬ | 310 | 40 | 250 | 180 | 60 |
| 4 | ▬▬▬ | 230 | 40 | 250 | 180 | 60 |
| 5 | ▬▬▬ | 315 | 40 | 250 | 180 | 60 |
| 6 | ▬▬▬ | 285 | 40 | 250 | 180 | 60 |
| 7 | ▬▬▬ | 285 | 40 | 250 | 180 | 60 |
| 8 | ▬▬▬ | 0 | 40 | 280 | 180 | 60 |

After finishing electrostatic spraying on the spraying line, enter the drying room to dry the car at high temperature. The 420 sampling points on the vehicle body are selected and a professional high precision coating thickness tester is used to measure the paint film thickness of each sampling point. The paint film thickness is shown in Figure 17. According to the measurement data, the coating thickness of automobile body is about 50 μm, and the difference between the maximum and minimum thickness of the coating is about 10 μm. In the actual production process of this brand of cars, the ideal coating thickness is 50 μm, and the maximum allowable deviation thickness is ±5 μm. It can be seen from the figure that a few measuring points exceed the upper limit of coating thickness, because these points are at the junction of the complicated surface. In general, the coating thickness basically meets the requirements, and the experimental results verify the effectiveness of the proposed method.

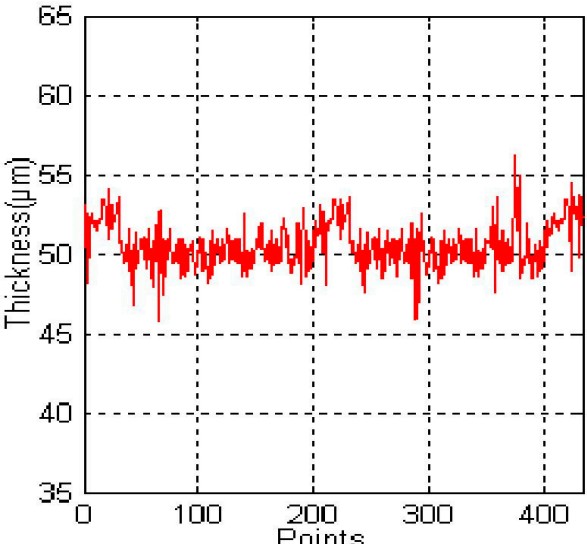

**Figure 17.** Oil film thickness data curve.

In this section, the trajectory optimization method of electrostatic spraying based on the surface modeling of point cloud slices is adopted in the spraying experiment, while the method of trajectory optimization based on CAD surface modeling is adopted in the reference [15–17] and [21–24]. Compared the two methods, the approach described in this section has two advantages: (1) In terms of spray efficiency, when spraying a large area of complex curved surface workpiece, since it is not necessary to triangulate the complicated surface, the point cloud section modeling method has high accuracy and speed, which can save about 15% of the system operation time for the same spraying workpiece; (2) In terms of spray effectiveness, after the spraying workpiece surface is segmented, if the spraying trajectory optimization method based on CAD curved surface modeling is adopted, the cumulative thickness uniformity of the coating at the junction between the pieces is poor, and some concave and convex points may even appear on the coating surface of the workpiece, which will affect the appearance of the automobile and other spraying workpiece; By using the trajectory optimization technology of electrostatic spraying based on point cloud slice, the overall coating thickness uniformity of the workpiece is relatively good, which can save about 20% of the coating for the same spraying workpiece.

## 6. Conclusions

At present, there are more and more complex spraying parts in industrial production, and traditional modeling and trajectory planning methods are gradually limited. With the upgrade of technology and process, consumers not only have high requirements for product quality, but also have higher pursuit for product appearance. When the product has the same quality, the appearance determines the competitiveness of the product. Aiming at the spraying problem of the workpiece with complicated shape in the process of industrial production, this paper first proposed a surface modeling method based on point cloud slice, and completed the surface modeling through data acquisition and data processing. Then the spraying trajectory planning method was proposed, which adopted the bias algorithm and interpolation algorithm to obtain the spraying robot's spatial trajectory. Finally, the spraying trajectory planning method based on point cloud slice technology and the trajectory model of electrostatic spraying fog drops are combined to carry out the spraying experiment on the automobile. The experimental results show that the method completely meet the requirements of coating thickness, the advantage of this method is to improve the coating uniformity and provide a solution for the modeling of large curvature workpiece or CAD free model. The disadvantage is

that the area segmentation adopts artificial method, adding artificial factors to a certain extent. If the automatic area segmentation can be achieved, the spraying trajectory can be automated.

**Author Contributions:** W.C. conceived and designed the experiments; W.C. and H.G. performed the experiments; Y.Z. and X.L. analyzed the data; X.L. contributed reagents/materials/ analyzed the data; L.W. contributed reagents/materials/analysis tools; X.L. wrote the paper. All authors have read and agreed to the published version of the manuscript.

**Funding:** This research is supported by the Project funded by Industrial Foresight Project of Zhenjiang City under grant GY2018018, "Six talent peaks" high-level talents projection Jiangsu Province under grant 2016GDZB021, the National Natural Science Foundation of China under grant 51809128, National Natural Science Foundation of China under grant 41906154.

**Conflicts of Interest:** The authors declare no conflict of interest.

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
