# Peer review of "Trajectory Planning for Spray Painting Robot Based on Point Cloud Slicing Technique"

_electronics, doi:10.3390/electronics9060908_

Round 1
Reviewer 1 Report
The article covers a wide range of topics and techniques related to spray painting with robots and point clouds. The authors did not manage to establish a clear contribution from the paper. The experiments are not in relation to the rest of the paper.
Main bottlenecks for the article:
- The introduction is weak, the literature overview in incomplete. Spray painting was one of the first application areas of the industrial robots back in 1967. Please look up: "Trallfa spray-paint robot". There are also scholars in Europe, USA who work on this field, without any reference to them. Like: "S. Moe, J. T. Gravdahl and K. Y. Pettersen, "Set-Based Control for Autonomous Spray Painting," in IEEE Transactions on Automation Science and Engineering, vol. 15, no. 4, pp. 1785-1796, Oct. 2018." https://ieeexplore.ieee.org/document/8305487
- Section 2 Spray Workpiece Surface Modeling, has minimal or no new contribution, beside description of the technology, which is already developed before.
- Section 2.3 Experiment on surface modeling of spraying workpiece, is a nice engineering work, but no clear summary or evaluation.
- Probably Section 3 is the main contribution of the article, but it is lacking details and highlighting the added contribution, to already existing knowledge.
- Section 5. Spraying Experiment, is total mess. No references to the previous sections. Neither any details about the experiments, which are relevant.
Reviewer 2 Report
The paper “Trajectory planning for Spray Painting Robot Based on Point Cloud Slicing Technique”, by Chen et al. proposes a spatial trajectory planning for spray painting robot based on point cloud slicing technology.
The paper is rather interesting but several issues need to be addressed. Also the topic seems to be too narrow for the readership of this Journal.
The abstract should be completely restructured. It is hard to understand what authors are investigating. Authors, after 1 or 2 lines introducing the problem, should quickly describe what they propose and their aim. a glimpse of the results should be reported. I short discussion and the take-home message should be provided.
Lines 52-54: I think that what is missing here is the introduction of the potentiality of biology inspirations and biorobotics. Authors should introduce here or in the discussion section the potentiality of these approaches. For instance, a spray painting robot could be improved by using goal-directed neural circuits such as central pattern generators. Also, concerning the trajectory, many organisms can be used as model to optimize the performances of this tasks in robotics (both for robot movements or for spraying materials).
This can also increase the general interest for the wide readership of this Journal.
Some useful examples that authors may appreciate to discuss and include in their work are:
A bioinspired autonomous swimming robot as a tool for studying goal-directed locomotion. Biological Cybernetics, 107(5), 513-527. (2013)
Biorobotics: Using robots to emulate and investigate agile locomotion. science, 346(6206), 196-203. (2014).
Impact of Aging and Cognitive Mechanisms on High-Speed Motor Activation Patterns: Evidence from an Orthoptera-Robot Interaction. IEEE Transactions on Medical Robotics and Bionics. (2020).
A review on animal–robot interaction: from bio-hybrid organisms to mixed societies. Biological Cybernetics, 113(3), 201-225. (2019).
In the model, it is not included a part concerning the coordinated work of different robotic arms painting the same object, that is crucial in industrial applications.
How do you address the overlapping issue? This would create different shades of colours on the object and a waste of material would also occur. This would produce a low quality manufacture and a non-cost effective work process.
A deep English revision is needed.
Round 2
Reviewer 1 Report
Thank you for the authors work on updating the manuscript.
The authors have replied to some extent of the reviewers, however there are still open questions:
For question: Section 2 Spray Workpiece Surface Modeling, has minimal or no new contribution, beside description of the technology, which is already developed before.
- The first part of section is unedited, and this the main part of the question here: results presented up to Section 2.2 are repetition, existing knowledge, without reference. E.g. Figure 3 is from Mingzhu Li, Zhangping Lu and Liqing Huang, "An approach to 3D shape blending using point cloud slicing," 2009 IEEE 10th International Conference on Computer-Aided Industrial Design & Conceptual Design, Wenzhou, 2009, pp. 919-922, doi: 10.1109/CAIDCD.2009.5375035. Some similarities in the text also.
- There is no point repeating results from previous work, without any contribution in the article.
For question: Section 5. Spraying Experiment, is total mess. No references to the previous sections. Neither any details about the experiments, which are relevant.
- The section looks much better. Can the authors describe/compare these results to any other spray-painting method? Where is the benefit with their proposal to compared to any other version?
Reviewer 2 Report
Authors provided good replies, but partially, to several points I raised. However, tìa coupled of them were overlooked.
I encourage authors to addres them more accurately in order to improve the quality of their interesting work
"Point 2: Lines 52-54: I think that what is missing here is the introduction of the potentiality of biology inspirations and biorobotics. Authors should introduce here or in the discussion section the potentiality of these approaches. For instance, a spray painting robot could be improved by using goal-directed neural circuits such as central pattern generators. Also, concerming the trajectory, many organisms can be used as model to optimize the performances of this tasks in robotics (both for robot movements or for spraying materials).
This can also increase the general interest for the wide readership of this Journal.
Some useful examples that authors may appreciate to discuss and include in their work are:
A bioinspired autonomous swimming robot as a tool for studying goal-directed locomotion. Biological Cybernetics, 107(5), 513-527. (2013)
Biorobotics: Using robots to emulate and investigate agile locomotion. science, 346(6206), 196- 203. (2014).
Impact of Aging and Cognitive Mechanisms on High-Speed Motor Activation Patterns: Evidence from an Orthoptera-Robot Interaction. IEEE Transactions on Medical Robotics and Bionics.(2020).DOI: 10.1109/TMRB.2020.2977003
A review on animal-robot interaction: from bio-hybrid organisms to mixed societies. Biological Cybernetics, 113(3), 201-225. (2019).
In the model, it is not included a part concerning the coordinated work of different robotic arms painting the same object, that is crucial in industrial applications".
Round 3
Reviewer 1 Report
The authors have replied to the reviewer's comments and the quality of the paper has increased.
Reviewer 2 Report
Authors addressed most of my comments.